# Characterization of Avian Influenza Viruses Detected in Kenyan Live Bird Markets and Wild Bird Habitats Reveal Genetically Diverse Subtypes and High Proportion of A(H9N2), 2018–2020

**DOI:** 10.3390/v16091417

**Published:** 2024-09-05

**Authors:** Peninah Munyua, Eric Osoro, Joyce Jones, George Njogu, Genyan Yang, Elizabeth Hunsperger, Christine M. Szablewski, Ruth Njoroge, Doris Marwanga, Harry Oyas, Ben Andagalu, Romona Ndanyi, Nancy Otieno, Vincent Obanda, Carolyne Nasimiyu, Obadiah Njagi, Juliana DaSilva, Yunho Jang, John Barnes, Gideon O. Emukule, Clayton O. Onyango, C. Todd Davis

**Affiliations:** 1Division of Global Health Protection, US Centers for Disease Control and Prevention, Village Market, Nairobi P.O. Box 606-00621, Kenya; 2Paul G. Allen School for Global Health-Kenya, Washington State University, Nairobi P.O. Box 72938-00200, Kenya; 3National Center for Immunization and Respiratory Diseases, US Centers for Disease Control and Prevention, 1600 Clifton Road, NE Atlanta, GA 30333, USA; 4Directorate of Veterinary Services, Ministry of Agriculture and Livestock Development, P.O. Box 29114, Kangemi, Nairobi 00625, Kenya; 5Influenza Division, US Centers for Disease Control and Prevention, Village Market, Nairobi P.O. Box 606-00621, Kenya; 6Centre for Global Health Research, Kenya Medical Research Institute, Kisumu P.O. Box 1578-40100, Kenya; 7Department of Veterinary Services, Kenya Wildlife Services, Nairobi P.O. Box 40241-00100, Kenya

**Keywords:** Africa, surveillance, poultry diseases, avian influenza, low pathogenic

## Abstract

Following the detection of highly pathogenic avian influenza (HPAI) virus in countries bordering Kenya to the west, we conducted surveillance among domestic and wild birds along the shores of Lake Victoria. In addition, between 2018 and 2020, we conducted surveillance among poultry and poultry workers in live bird markets and among wild migratory birds in various lakes that are resting sites during migration to assess introduction and circulation of avian influenza viruses in these populations. We tested 7464 specimens (oropharyngeal (OP) and cloacal specimens) from poultry and 6531 fresh fecal specimens from wild birds for influenza A viruses by real-time RT-PCR. Influenza was detected in 3.9% (n = 292) of specimens collected from poultry and 0.2% (n = 10) of fecal specimens from wild birds. On hemagglutinin subtyping, most of the influenza A positives from poultry (274/292, 93.8%) were H9. Of 34 H9 specimens randomly selected for further subtyping, all were H9N2. On phylogenetic analysis, these viruses were genetically similar to other H9 viruses detected in East Africa. Only two of the ten influenza A-positive specimens from the wild bird fecal specimens were successfully subtyped; sequencing analysis of one specimen collected in 2018 was identified as a low-pathogenicity avian influenza H5N2 virus of the Eurasian lineage, and the second specimen, collected in 2020, was subtyped as H11. A total of 18 OP and nasal specimens from poultry workers with acute respiratory illness (12%) were collected; none were positive for influenza A virus. We observed significant circulation of H9N2 influenza viruses in poultry in live bird markets in Kenya. During the same period, low-pathogenic H5N2 virus was detected in a fecal specimen collected in a site hosting a variety of migratory and resident birds. Although HPAI H5N8 was not detected in this survey, these results highlight the potential for the introduction and establishment of highly pathogenic avian influenza viruses in poultry populations and the associated risk of spillover to human populations.

## 1. Introduction

Influenza A viruses are members of the *Orthomyxoviridae* family and contain a segmented, negative-sense RNA genome. They are commonly described by their combinations of two surface proteins, hemagglutinin (H; H1 to H18) and neuraminidase (N; N1 to N11). While the viruses replicate in numerous terrestrial avian and mammalian species, wild aquatic birds are considered the natural reservoirs of avian influenza viruses (AIVs) [1].

Globally, there has been a substantial increase in detections of diverse highly pathogenic avian influenza (HPAI) and low-pathogenicity avian influenza (LPAI) viruses in wild birds and poultry in the past two decades, including in Africa [2,3]. The evolution of new strains and the dispersal and establishment of these strains in new areas across the world have been associated with seasonal wild bird migration along flyways [2,4]. One such migratory pathway, the East Africa–West Asia flyway, encompasses portions of southern and eastern Africa [4]. Likewise, poultry trade and, specifically, live bird markets (LBM) are recognized as key nodes for the transmission of influenza A viruses and act as key sites to monitor for the introduction and circulation of AIVs in poultry populations.

Globally, human infections with different AIV H5, H7, and H9 influenza A subtypes with varying N proteins have occurred, commonly as spillover events during AIV outbreaks in poultry [5,6]. The earliest human infections with H9N2 were reported in China in 1998 [6]. By 2019, through ongoing targeted surveillance, 59 H9N2 human cases had been reported in China, Hongkong, Egypt, Bangladesh, Pakistan, and Oman. The majority of AIV human cases have reported exposure to poultry, underscoring an ongoing public health risk given the intrinsic potential for influenza virus reassortment [5,6,7,8]. In 2021–2023, the WHO reported 20 human cases of HPAI H5N1 virus [9]. All but one case in India had confirmed exposure to poultry [10].

Low-pathogenicity avian influenza (LPAI) H9N2 viruses are the most prevalent subtype of AIVs in poultry globally and have been reported throughout Asia, the Middle East, Europe, and Africa, including in Kenya [6,7,11,12]. Since introduction in 1966, H9N2 has been frequently observed with high prevalence rates, exceeding 10% among poultry in LBMs in Vietnam, China, and Egypt [6]. H5N2 viruses are less prevalent in Africa, and while HPAI H5N2 outbreaks have been reported in farmed ostriches in South Africa, and the LPAI H5N2 viruses have been reported in West Africa, the H5N2 viruses have not been reported in Eastern Africa [13,14]. Between 2016 and 2018, an Asian-origin HPAI H5N8 rapidly spread in a variety of avian species, including wild birds and poultry, in sub-Saharan Africa [11,15,16,17]. The pan-African spread of H5N8 demonstrated the potential for large-scale transmission of AIVs in different ecosystems across the continent, highlighting knowledge gaps on the contribution of environmental conditions on AIV transmission and underscoring the need for more robust continental surveillance for monitoring and mitigation measures [15]. In January 2017, Uganda confirmed infection with HPAI H5N8 among wild birds in Lutembe Bay along the shores of Lake Victoria and in poultry in two districts of its southeastern region [16]. In response to the outbreak, risk assessment considered the spread of H5N8 across Eastern African countries as likely [16]. On phylogenetic analysis, the H5N8 viruses in the Uganda outbreak clustered with viruses collected in 2017 in the Democratic Republic of the Congo and in West Africa, suggesting migratory birds could have played a role in transmission in the region. 

In Kenya, AIVs surveillance among poultry and wild birds has not been routinely conducted. Surveillance conducted from 2009–2011 detected AIV in only 0.8% of chickens [18], and limited LBM sampling in 2017 reported 5.7% AIV positivity [12]. In response to the 2018 outbreaks of H5N8 virus in Uganda, we initiated surveillance for AIVs among poultry traded in LBMs and among wild birds to assess possible introduction and circulation of AIVs in these populations in Kenya. In addition, given the potential of human infections with AIVs, we conducted surveillance for influenza viruses among poultry workers in the LBMs.

## 2. Materials and Methods

### 2.1. LBM Surveillance

#### 2.1.1. Site Selection

We targeted LBMs for surveillance of poultry and poultry workers from March 2019 through February 2020 within five counties of Kenya, including Nairobi, Nakuru, Kisumu, Kiambu, and Busia. Kisumu and Busia were selected because of their proximity to poultry trade flow with Uganda, where a HPAI H5N8 outbreak was previously reported [16]. Nakuru and Nairobi were selected to represent areas that receive poultry from different parts of Kenya. The market in Kiambu was selected because it primarily traded in non-chicken poultry. Overall, seven LBMs were selected, including three in Nairobi and one market in each of the other counties.

#### 2.1.2. Sample Size Estimation 

We calculated the number of poultry specimens collected based on the estimated seroprevalence of 0.8% (95% confidence interval [CI] 0.6–1.1) [18] reported among Kenyan poultry in LBMs in 2009 with an error margin of 1.3% at the 95% confidence level using formulas described by Thompson [19]. A design effect of 1.5 was applied to account for clustering resulting in a sample size of 5380.

#### 2.1.3. Specimen Collection

Poultry: LBMs were visited monthly for the duration of the survey. Up to 20 stalls were selected in each market and 3–5 birds sampled in each stall. In Nyambari market in Kiambu County, where poultry were not caged, approximately 30% of the birds available at the market on any day of sample collection were sampled by species. For each sampled poultry, one oropharyngeal (OP) and one cloacal (CL) specimen were collected.

Poultry workers: All poultry workers aged >13 years were eligible for enrollment after providing consent/assent. A questionnaire collecting demographic characteristics of the enrolled participant was completed during the first contact. On subsequent monthly visits, a brief questionnaire on respiratory illness during the visit or the preceding 14 days was administered. Those who met the case definition for acute respiratory infection (ARI), defined as history of fever, cough, or red eyes, and with onset within the last seven days, had an oropharyngeal (OP) and nasal (NS) specimens collected for influenza testing. Sample size calculations were not performed for poultry worker surveillance.

### 2.2. Wild Bird Surveillance

#### 2.2.1. Site Selection

During January 2018, we collected freshly voided fecal specimens from wild birds on the shores of Lake Victoria, along its edges from Homabay to Busia County. From February to March 2020, sampling was conducted along the shorelines of Lake Bogoria, Lake Elementaita, Lake Nakuru, Lake Naivasha, and Lake Victoria (Figure 1).

#### 2.2.2. Sample Size Estimation

Prevalence of AIVs among wild birds in Kenya is not known, but prevalence of AIVs in fecal specimen pools was reported at 2.3%, where 12/504 pools (of five specimens each) tested positive (and 3.7% in fecal specimens collected from fresh water lakes) [20]. Using the lower prevalence and targeting surveillance strategies to detect at least 15 positive pools, we set out to collect 3260 specimens to detect a prevalence as low as 1.2% at a 95% confidence level.

#### 2.2.3. Sample Collection

At each site, an ornithologist on the team recorded data on the bird species observed including the dominant species, their migratory status (Palearctic, Afrotropical, or local), presence of dead birds, and any wild bird interaction with humans or poultry. Geographic coordinates for each site were also recorded.

Fresh environmental fecal specimens were collected using plastic-shafted polyester-tipped swabs and placed individually in cryovials containing 2 mL of freshly prepared viral transport media (VTM) containing bovine serum albumin and veal infusion broth supplemented with amphotericin B and gentamycin https://iris.who.int/handle/10665/68026, accessed on 3 July 2024.

### 2.3. Virus Testing and Sequencing Analysis

Specimens from LBMs were labeled and maintained at 2–8 °C upon collection and subsequently frozen at −80 °C at the U.S. Centers for Disease Control and Prevention (CDC)-supported Kenya Medical Research Institute (KEMRI) Center for Global Health Research (CGHR) laboratory in Kisumu until testing. Wild bird fecal specimens were placed in cool boxes after collection and then frozen in liquid nitrogen within four hours of collection and subsequently frozen at −80 °C until testing. Due to the high cost of reagents, a subset (72%) of collected specimens were screened for influenza A. Poultry worker specimens were maintained at 2–8 °C upon collection and subsequently frozen at −80 °C at the CDC-supported KEMRI-CGHR laboratory in Kisumu until testing. 

Specimens were pooled in groups of five according to species and site of collection prior to RNA extraction. Pooled specimens were tested for influenza A virus RNA by real-time reverse transcription-polymerase chain reaction (RT-PCR) using primers and probes that target the matrix gene of influenza A viruses [21]). The cut-off for positivity was a cycle threshold (CT) value ≤ 40. Specimens from any positive pool were retested individually. Viral subtype was determined using primers and probes targeting the hemagglutinin genes of three avian influenza A subtypes (H5, H7, and H9). Human specimens were screened individually for influenza A virus by RT-PCR. 

Full genome sequencing was performed at the Influenza Division laboratories (CDC, Atlanta) on a random sample of positive LBM specimens (n = 34) and all wild bird specimens (n = 2) that tested positive for influenza A virus and had a CT value of <30. Total RNA was extracted from 100 μL of sample using the MagNA Pure 96 DNA and Viral NA Small Volume Kit (catalog no. 06542588001; Roche Diagnostics, Indianapolis, Indiana) according to the manufacturer’s protocol. Purified RNA was eluted into 50 μL nuclease-free water and screened for influenza A virus and Newcastle disease virus (NDV) using TaqMan real-time RT-PCR assays) [22,23]. Full genome sequencing was performed using Illumina MiSeq technology, as previously described [24]. Sequences were deposited in Global Initiative on Sharing All Influenza Data (GISAID) (GISAID accession numbers EPI_ISL_18856481–EPI_ISL_18856545.).

HA sequences from Kenyan H9N2 isolates were used to construct a phylogenetic tree. Data were aligned using multiple sequence comparison by log-expectation (MUSCLE) [25], and sequences were trimmed to the beginning of the mature HA1 protein gene sequence using BioEdit v7.0. The neighbor-joining method [26] with the Jukes–Cantor model in the Mega 7.0 software package [27] was used to construct the final tree. The percentage of replicate trees in which the associated taxa clustered together was measured by the bootstrap test (1000 replicates), and bootstrap values are shown next to the branches. Nucleotide sequence alignments used for the HA trees were translated to amino acid protein sequences for comparison to the closest A(H9N2) World Health Organization (WHO) candidate vaccine virus (CVV), A/Oman/2747/2019. Amino acid changes are annotated on each branch of the tree.

### 2.4. Antigenic Characterization of AIVs

Specimens received at the CDC Influenza Division laboratories were inoculated into 10-day old embryonated eggs, incubated for 48 h at 37 °C, and harvested [28]. The antigenic relatedness of the six isolates was determined using the hemagglutination inhibition (HI) assay. The ferret antisera used in the HI test were created by inoculating naïve ferrets intranasally with selected reference virus using a 50% egg infectious dose (EID50) of 10^6^. Antisera were harvested 14 days post infection. Each ferret antiserum was treated with a receptor-destroying enzyme (RDE: Denka Seiken) at a 1:4 dilution for 18 h at 37 °C and further diluted with physiological saline for a total dilution of 1:10. The diluted antiserum was adsorbed with packed turkey red blood cells. Antiserum was serially diluted 2-fold in a 96-well v-bottom plate, and all antigens were standardized to 8 hemagglutination units/50 µL. Antisera and antigens were incubated together for 30 min at room temperature. Standardization and hemagglutination inhibition were determined using turkey red blood cells (0.5%) [28].

## 3. Results

Between March 2019 and February 2020, 10,340 poultry specimens were collected from seven LBMs in five counties in Kenya (Table 1). Most of the specimens were collected from chickens (8176; 79%), followed by ducks/geese (1166; 11%), turkeys (590; 6%), pigeons (270; 3%), domesticated guinea fowl (114; 1%), and doves (24; 0.2%). The median stay of poultry at the market was 1 day for chickens [interquartile range (IQR), 1–2 days], 9 days [IQR 5–17 days] for guinea fowls, 12 days [IQR 10–21 days] for pigeons, 12 days [IQR 9–22 days] for doves, 16 days [IQR 10–21 days] for ducks/geese, and 17 days [IQR 10–21 days] for turkeys.

During the LBM surveillance period, 155 poultry workers at the market were enrolled for follow-up from Burma (37; 24%), Busia (19; 12%), Kariokor (14; 9%), Kawangware (16; 10%), Kisumu (28; 18%), Nakuru (23; 15%), and Nyambari (18; 12%). The median age was 38 years [IQR 30–46 years], and 91 (59%) were males. Three quarters of the participants (n = 120) were present during all 12 follow-up visits. A total of 18 OP/NS swabs from poultry workers with ARI (12%) were collected; none were positive for influenza A virus.

Across five lakes in Kenya, a total of 6531 fecal specimens from wild birds were collected, including 3351 (51.3%) in January 2018 (Lake Victoria only) and 3180 (48.7%) in February–March of 2020 (Figure 1, Table 1). A total of 580 different bird species were observed in the sampling sites, including 26% (n = 149) Palearctic migrants, 19% (n = 109) Afrotropical migrants, and 56% (n = 322) local types (Appendix A).

### 3.1. Detection of Influenza A Virus in LBM Poultry Specimens

From the specimens collected at LBMs (n = 10,340), we randomly selected and tested 7464 (72.2%), including 3737 cloacal specimens and 3727 OP specimens in 1507 pools. A total of 135/1507 (9.0%) pools were positive for influenza A, segregated as 19/754 (2.5%) cloacal and 116/753 (15.4%) OP pools. On individual specimen testing, overall, 292 (3.9%) specimens were positive for influenza A, including 246/3727 (6.6%) OP specimens and 46/3737 (1.2%) cloacal specimens. Chickens most frequently tested positive for influenza A viruses (5.0%), followed by turkeys (0.3%) and ducks/geese (0.2%) (Table 2). None of the specimens collected from pigeons, guinea fowl, and doves tested positive for influenza A. On hemagglutinin subtyping, most of the influenza A-positive specimens (274/292, 93.8%) were H9. We randomly selected 34 H9 specimens for further subtyping, and all were H9N2.

By region, influenza A was most commonly detected among specimens collected in Kawangware (17.5%), followed by Burma (11.1%) and Busia (0.7%). None of the specimens collected from the Nakuru market were positive (Table 2). Influenza A was detected in 12 of 13 months during LBM surveillance, with detection most common during April 2019 and least common during December 2019 (range 0–8.2%) (Figure 2). 

### 3.2. Detection of Influenza A Virus in Wild Bird Fecal Specimens

A total of 6531 specimens (in 1307 pools of five) were tested, and 10/1307 pools (0.8%) were positive for influenza A. Of individually tested specimens, 10/6531 (0.2%; 95% CI) 0.1–0.3) were positive for influenza A, including 1/3351 (0.0%; 95% CI 0.0–0.2) collected in 2018 and 9/3180 (0.3%; 95% CI 0.1–0.5) collected in 2020. Among specimens collected in 2020, influenza A positivity was slightly more frequent in specimens from Lake Nakuru (0.7%; 4/560), followed by 0.4% (2/484) from Lake Elementaita and 0.3% (3/904) from Lake Naivasha; fecal specimens collected from Lake Bogoria and Lake Victoria basin tested negative.

### 3.3. Phylogenetic Analyses of the H9N2 Viruses from LBM Poultry

We obtained codon complete nucleotide sequences of all eight segments of 34 AIV isolates, including 32 from chickens and 2 from turkeys. Phylogenetic analyses of the HA sequences showed that the isolates belonged to the G1 lineage of H9N2 viruses (Figure 3). Viruses from this study clustered with sequences of specimens collected in Uganda in 2017. On the same phylogenetic tree, sequences from West Africa and North Africa clustered separately from the East Africa viruses, indicating evidence of local and regional circulation among poultry. Moreover, amino acid differences relative to the closest CVV in the G1 linage, A/Oman/2747/2019, indicate multiple conserved changes in the HA1 portion of the sequenced viruses from Kenya. Viruses sequenced from this study had between 9–13 amino acid changes compared to the reference CVV, of which several were identified at putative antigenic or receptor binding sites. All the viruses from this study had amino acid changes G150L at antigenic site B. Other mutations of the antigenic sites included N148S at antigenic site B and N161S and R162Q at antigenic site D (Appendix A). Of note, viruses collected from turkeys had changes of either L216R or Q compared to viruses from chickens, indicating likely species-specific receptor binding among terrestrial poultry.

### 3.4. Phylogenetic Analysis of the H5N2 Virus from Wild Bird Fecal Specimens

One influenza A-positive fecal specimen collected from the Lake Victoria basin in 2018 was H5-positive and, upon phylogenetic analysis, determined to be LPAI A(H5N2) virus of the Eurasian lineage. We obtained whole-genome codon complete sequences of all eight segments of the A/environment/Kenya/Z201801195/2018 A(H5N2) virus. Additionally, one of the nine influenza A-positive wild bird fecal specimens collected in 2020 was successfully sequenced and was subtyped as an A(H11) virus. Only the HA could be sequenced.

### 3.5. Antigenic Analysis Results of H9N2 Poultry Specimens

A panel of genetically related A(H9N2) viruses and CVVs were compared by HI assay to six A(H9N2) viruses from Kenya to determine antigenic relatedness. The panel of reference viruses and ferret antisera included three G1-lineage WHO CVVs: A/Hong Kong/1073/1999, IDCDC-RG31 (A/Bangladesh/0994/2011-like), and IDCDC-RG66A (A/Oman/2747/2019-like), a more recent CVV of the G1 lineage [29].

The viruses isolated from Kenya were A/chicken/Kenya/BU-CO-0688/2019, A/chicken/Kenya/BM-CO-0136/2019, A/chicken/Kenya/KN-CO-0454/2019, A/chicken/Kenya/BM-CO-0756/2019, A/chicken/Kenya/BM-CO-0808/2019, and A/chicken/Kenya/BM-CO-0010/2020 H9N2. Amino acid comparison of the hemagglutinin protein of the six isolates versus the closest genetically related WHO CVV IDCDC-RG66A (A/Oman/2747/2019-like) showed between 11 and 13 amino acid changes (Appendix A). One amino acid change (N148S) in putative antigenic site B was identified in multiple H9N2 viruses and two that were selected for virus isolation and HI testing: A/chicken/Kenya/BM-CO-0756/2019 and A/chicken/Kenya/BM-CO-0808/2019. Amino acid 148S in combination with 216L in antigenic site B was shown to increase α2-6 binding [30].

Antigenic analysis showed that the six virus isolates were inhibited within four-fold of the homologous titers of the most recent G1-lineage WHO CVV, namely IDCDC-RG66A (A/Oman/2747/2019-like). A CVV previously selected in the G1 lineage, IDCDC-RG31 (A/Bangladesh/0994/2011-like), also inhibited viruses within four-fold of the homologous virus titer (Table 3), indicating that the identified amino acid differences did not result in substantial antigenic variation relative to the G1-lineage CVVs.

## 4. Discussion

Using multiple strategies and locations to monitor and identify avian influenza viruses in Kenya, we identified avian influenza viruses among traded poultry and wild birds, including two low-pathogenicity viruses, H9N2 and H5N2. We found influenza A positivity in 3.9% of specimens collected from poultry traded in Kenyan LBMs between 2019 and 2020. On phylogenetic analysis, these viruses were genetically similar to other H9 viruses detected in East Africa. Concurrently, we found a low prevalence of circulation of LPAI H5N2 in wild bird populations. While there was no apparent transmission of these viruses to persons working in the LBMs during this monitoring period, related viruses belonging to the G1 lineage have resulted in human infections in Africa and elsewhere, underscoring the public health threat and need for continuous surveillance [6,7,31]

The observed prevalence in poultry represents a five-fold increase in influenza A virus detection compared with similar surveillance conducted a decade earlier (0.8% positivity during 2009–2011) [18]. Virus detection was five times more frequent among tested oropharyngeal specimens (6.6%) compared to cloacal specimens (1.2%), a common observation among poultry [18,32]. H9N2 viruses were detected in chicken and turkey specimens during this period, and detection was variable by the month of collection (0.0–8.2%) and by market. Two of three LBMs in Nairobi that primarily traded indigenous poultry sourced from different counties had significantly higher influenza A virus positivity compared to all the LBMs. Additionally, the detection of AIVs in different poultry species that come into contact with each other at LBMs for extended periods, up to 17 days, could potentially facilitate cross-species transmission. This suggests widespread transmission of H9N2 viruses in poultry populations across Kenya. High prevalence of H9N2 viruses has been reported in many countries in Asia, the Middle East, and Africa [6]. The overall H9N2 prevalence observed was higher than previously reported (5.6%) in 2018 in Kenya [12].

Phylogenetic analyses showed that all 34 H9N2 viruses belonged to the G1 lineage, with minimal genetic variation among the cluster detected in Kenya, suggesting circulation of viruses from a single lineage. As previously shown [12], viruses from this study in Kenya also clustered with those detected in Uganda, a neighboring country, further supporting that the outbreaks in East Africa were likely from a single lineage. This indicates that despite the contemporaneous circulation of virus clusters identified in West, North, and East Africa, distinct clusters were delineated, implying that H9N2 viruses in East Africa may have evolved independently with no intermingling of virus gene pools. Analysis of HA protein sequences identified mutations at several antigenic sites (125, 148, 150, 161, 162, 178, and 216) compared to the IDCDC-RG66A CVV (A/Oman/2747/2019-like). Despite these changes, the viruses isolated in this study remained antigenically similar to G1-lineage CVVs recommended for vaccine development by the WHO.

We found a low influenza prevalence among wild bird populations in Kenya. This prevalence was lower than that observed during the 2008 study (2.3%) in Kenya that detected H4N6 [20]. On subtyping, we detected LPAI H5N2 in one fecal specimen. Phylogenetic analysis of the hemagglutinin (HA) gene of isolates from wild bird fecal specimen clustered with viruses from the Eurasian lineage. The LPAI H5N2 virus was collected around the Lake Victoria basin, an important bird resting area that hosts a variety of Palearctic and Afro-tropical migrants in addition to resident bird species. LPAI H5N2 viruses in migratory birds and poultry have been documented around the globe, including in Nigeria in 2008 among wild birds with Eurasian migratory patterns, implicating introduction of the virus to wild birds in African wetlands. Notably, during the 2016–2018 period, there were multiple independent introductions of HPAI H5N8 to different countries on the African continent by migratory birds, and sporadic spillover to poultry populations, including the first ever recorded introduction of HPAI H5N8 in East Africa, was recorded [2,15,16,32]. While LPAI H5N2 was not reported in poultry in this region over that period, it is likely that the virus was introduced during the same period.

These findings have several limitations. Because of limited resources, sampling among wild birds was conducted in only two time periods and in only one site (Lake Victoria basin) in 2018. Hence, we may have failed to detect some AIV introductions among wild bird populations. Similarly, LBM surveillance was limited to a certain number of counties and spanned only one year, which may not provide a comprehensive representation of all AIVs circulating in poultry populations in Kenya. However, the findings of this study highlight the importance of continued AIV surveillance in domestic and wild bird populations.

Since 2020, there have been reports of an increase in detection of AIVs in Europe, Asia, and North America. However, detections in the African continent have been limited to certain countries in southern and western Africa. Due to insufficient surveillance of AIVs in poultry and wild bird populations in most African countries, our findings along with recent findings indicating rapid annual expansion of HPAI H5N8 and LPAI H9N2 on the African continent demonstrate significant risk of virus introduction from outside of the continent, likely due to persistence in local poultry and wild bird populations, and potential spillover to domestic birds and humans. Adoption of suitable biosecurity measures in LBMs could mitigate cross-species transmission and opportunity for virus reassortments. Strategies such as systematic LBM surveillance and targeted surveillance for influenza viruses in wild bird populations aligned to the seasonal migratory patterns could achieve early detection of virus incursion, providing the opportunity for intervention measures to mitigate socioeconomic and public health impacts caused by outbreaks of HPAI viruses.

## Figures and Tables

**Figure 1 viruses-16-01417-f001:**
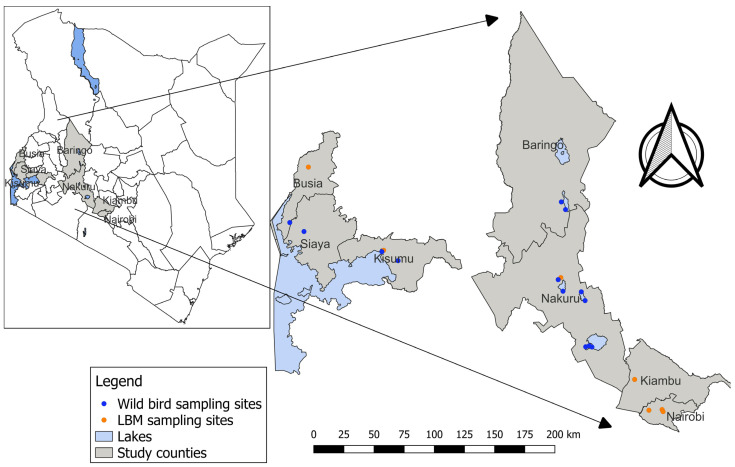
Map of Kenya showing sampling sites for avian influenza viruses in live bird markets (LBM) and wild birds.

**Figure 2 viruses-16-01417-f002:**
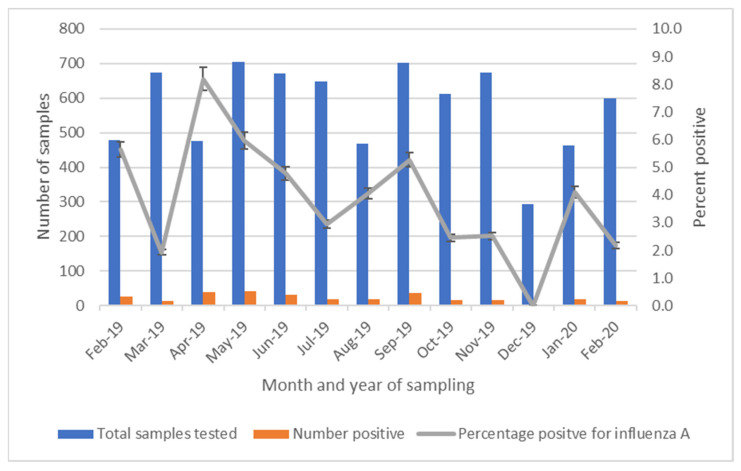
Live bird market surveillance specimens collected and percent positivity by month of collection, Kenya, 2019–2020.

**Figure 3 viruses-16-01417-f003:**
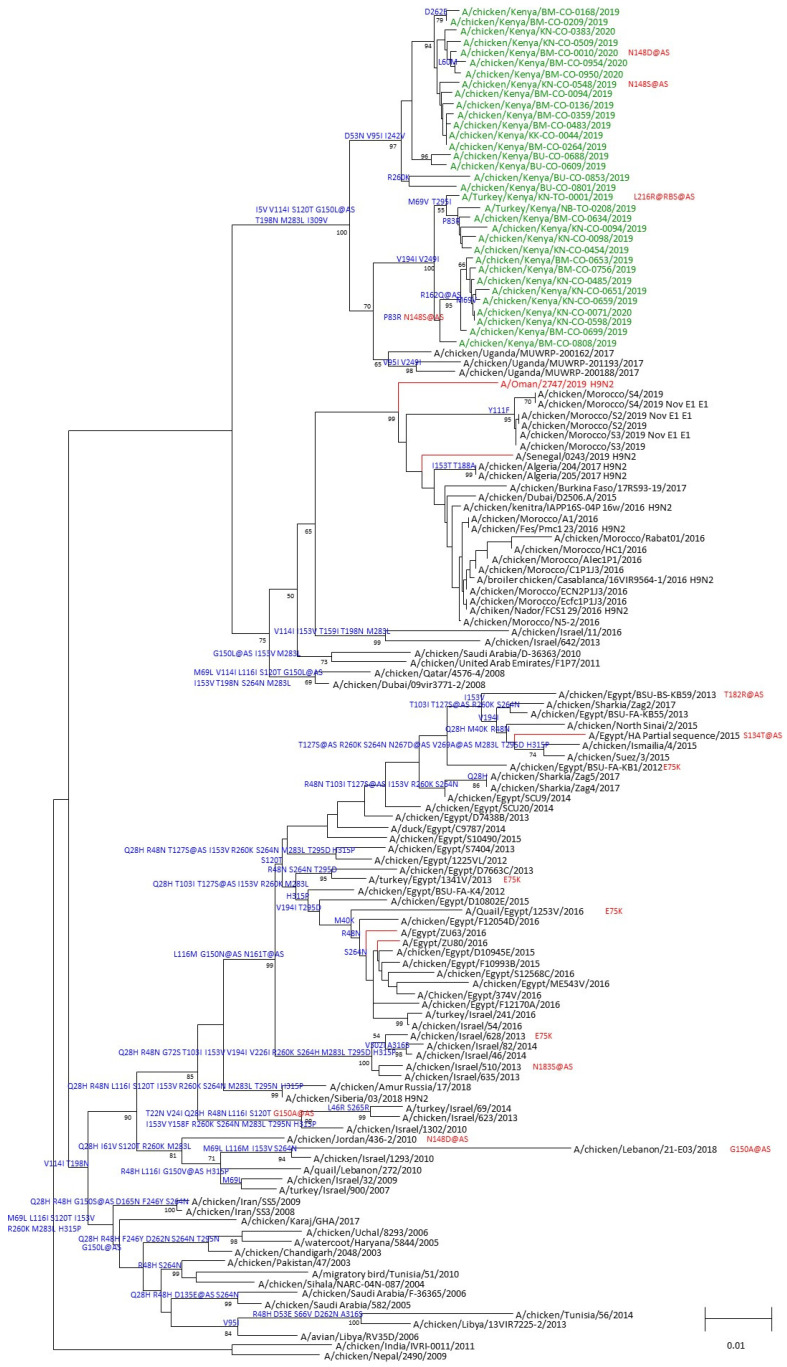
Phylogenetic tree of the hemagglutinin genes of representative G1-lineage influenza H9N2 viruses. Specimens sequenced for this study are in green. The nearest candidate vaccine virus is shown in red. Amino acid changes relative to the nearest candidate vaccine virus are shown on each branch. Bootstrap support for each node is shown representing 1000 replicates. Mutations found in the H5N1 genetic changes inventory are shown in red. Mutations at putative antigenic site is denoted by @AS. Mutations at receptor binding sites are denoted by @RBS.

**Table 1 viruses-16-01417-t001:** Poultry and wild bird sample collections in Kenya, 2018–2020.

Sampling Location	County	Main Bird Type Sampled	Total Specimens Collected n (%)
Burma market	Nairobi	Chicken	1928 (18.6)
Busia market	Busia	Chicken	1496 (14.5)
Kariokor market	Nairobi	Chicken	490 (4.7)
Kawangware market	Nairobi	Chicken	973 (9.4)
Kisumu market	Kisumu	Chicken	1710 (16.5)
Nakuru market	Nakuru	Chicken	1633 (15.8)
Nyambari market	Kiambu	Various *	2110 (20.4)
Total			10,340
Wild bird fecal sampling			
Lake Bogoria	Baringo	Wild birds’ fecal specimens	305 (4.7)
Lake Elementaita	Nakuru	Wild birds’ fecal specimens	484 (7.4)
Lake Nakuru	Nakuru	Wild birds’ fecal specimens	560 (8.6)
Lake Naivasha	Nakuru	Wild birds’ fecal specimens	904 (13.8)
Lake Victoria basin	Various ^†^	Wild birds’ fecal specimens	4278 (65.5) ^ᵦ^
Total			6531

* Poultry types sampled included 54% duck/geese, 27% turkeys, 13% pigeons, and 5% guinea fowls. ^†^ Lake Victoria basin spans across Busia, Kisumu, Homabay, and Siaya counties. ^ᵦ^ A total of 3351 (51.3%) specimens were collected in 2018.

**Table 2 viruses-16-01417-t002:** Real-time RT-PCR test results by specimen type, poultry type, and market, Kenya, 2018–2020.

Characteristic	Specimen Type	Total Specimens Tested	Influenza A-Positive n (%)	Influenza A Percent Positive 95% CI
Overall	OP	3727	246 (6.6)	5.8–7.4
	CL	3737	46 (1.2)	0.9–1.6
	Total	7464	292 (3.9)	3.5–4.3
Poultry species				
Chicken	OP	2928	243 (8.3)	7.3–9.3
	CL	2889	46 (1.6)	1.1–2.0
	Total	5817	289 (5.0)	4.4–5.5
Turkey	OP	291	2 (0.7)	0.1–2.5
	CL	294	0 (0.0)	-
	Total	585	2 (0.3)	0.04–1.2
Duck/Geese	OP	305	1 (0.3)	0.01–1.8
	CL	305	0 (0.0)	-
	Total	610	1 (0.2)	0.004–0.9
Pigeon	OP	136	0 (0.0)	-
	CL	136	0 (0.0)	-
	Total	272	0 (0.0)	-
Domestic guinea fowl	OP	55	0 (0.0)	-
	CL	55	0 (0.0)	-
	Total	110	0 (0.0)	-
Domestic dove	OP	12	0 (0.0)	-
	CL	12	0 (0.0)	-
	Total	24	0 (0.0)	-
Live Bird Market Name [county]				
Kawangware [Nairobi]		719	126 (17.5)	14.7–20.3
Burma [Nairobi]		1393	155 (11.1)	9.5–12.8
Busia [Busia]		1033	7 (0.7)	0.3–1.4
Kariokor [Nairobi]		355	2 (0.6)	0.1–2.0
Nyambari [Kiambu]		1547	1 (0.1)	0.002–0.3
Kisumu [Kisumu]		1224	1 (0.1)	0.002–0.4
Nakuru [Nakuru]		1193	0 (0.0)	-

RT-PCR: reverse transcription-polymerase chain reaction; OP: oropharyngeal; CL: cloacal.

**Table 3 viruses-16-01417-t003:** Hemagglutination inhibition reactions of influenza A/H9N2 G1-lineage viruses.

	Hemagglutination Inhibition Titers ^a^				
	HK/1999	BG/0994	RG31	Oman	RG66A
Reference Antigens ^b^					
A/Hong Kong/1073/1999	1280	20	20	40	10
A/Bangladesh/0994/2011	160	2560	1280	640	1280
IDCDC-RG31 (A/Bangladesh/0994/2011-like)	160	2560	1280	320	1280
A/Oman/2747/2019	160	640	160	1280	640
IDCDC-RG66A (A/Oman/2747/2019-like)	160	1280	320	1280	2560
Test Antigens					
A/chicken/Kenya/BU-CO-0688/2019	80	1280	320	1280	1280
A/chicken/Kenya/BM-CO-0136/2019	80	2560	1280	1280	1280
A/chicken/Kenya/KN-CO-0454/2019	80	1280	640	640	640
A/chicken/Kenya/BM-CO-0756/2019	160	1280	1280	1280	1280
A/chicken/Kenya/BM-CO-0808/2019	160	2560	1280	1280	1280
A/chicken/Kenya/BM-CO-0010/2020	80	640	320	640	640

**^a^** The number in bold is the homologous titer. ^b^ HK/1999: A/Hong Kong/1073/1999; BG/0994: A/Bangladesh/0994/2011; RG31: IDCDC-RG31 (A/Bangladesh/0994/2011-like); Oman: A/Oman/2747/2019; RG66A: IDCDC-RG66A (A/Oman/2747/2019-like).

## Data Availability

All data underlying this article are available in the article.

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
