# Peer review of "Characterization of Avian Influenza Viruses Detected in Kenyan Live Bird Markets and Wild Bird Habitats Reveal Genetically Diverse Subtypes and High Proportion of A(H9N2), 2018–2020"

_viruses, 2024, doi:10.3390/v16091417_

Round 1

Reviewer 1 Report

Comments and Suggestions for Authors

Reviewer comments on the manuscript entitled: “Characterization of Avian Influenza Viruses Detected in Kenyan Live Bird Markets and Wild Bird Habitats Reveal Genetically Diverse Subtypes and High Proportion of A(H9N2), 2018-2020” Submitted by Munyua and colleagues for consideration for publication in Viruses, section, Animal Viruses.

In this manuscript, the authors tested 7,464 samples from poultry (oropharyngeal and cloacal swabs) and 6,531 fecal samples from wild birds collected between 2018-2020 in Kenyan live bird markets and wild bird habitats, particularly around Lake Victoria. Using PCR, 3.9% (292) of poultry specimens and 0.2% (10) of wild bird specimens tested positive for IAV. 93.8% (274/292) of influenza-positive poultry specimens were H9N2, which were genetically similar to other H9 viruses detected in East Africa. In wild birds, two viruses were identified; H5N2 (2018, Eurasian lineage), and the other was H11 (2020). In addition, none of the Poultry Workers (n= 18) with flu-like illness were positive for IAV. These results indicated significant circulation of potentially zoonotic H9N2 in poultry; low pathogenic H5N2 found in wild birds; no HPAI H5N8 detected.

The manuscript addresses a critical gap in understanding the circulation of AIVs, particularly the H9N2 subtype, in East Africa. This is particularly important given the potential for these viruses to spill over into human populations, as well as the risks associated with the introduction of HPAI strains particularly the currently panzootic H5Nx viruses.

This research significantly contributes to the subject area by providing the first detailed characterization of AIVs in Kenyan poultry and wild birds over a multi-year period. The identification of H9N2 as the predominant subtype in poultry, along with the detection of H5N2 and H11 in wild birds, adds valuable data to the global understanding of AIV distribution and evolution, particularly in under-studied regions like East Africa.

The methodology used is robust, employing real-time RT-PCR and sequencing for virus detection and characterization. The conclusions are consistent with the evidence and arguments presented. The findings are well-supported by the data and highlight important implications for public health and avian influenza management in the region. The references used in the manuscript are appropriate and relevant to the study. They provide a solid background and context for the research, drawing on key studies in the field of avian influenza. The tables and figures are clear, well-organized, and effectively present the data. 

Comments on the Quality of English Language

No serious comments

Author Response

There is not specific comment from the reviewer that needs to be addressed. The reviewer notes that minor editing of English language required. We have read through the ms and made edits.

We thank the reviewer for critical review and are particularly grateful for your endorsement of our paper as well-written and useful for publication. Your positive evaluation reinforces our belief in the significance and relevance of our research findings

Reviewer 2 Report

Comments and Suggestions for Authors

The study presented is noteworthy, but suffers from the following critical issues:
- I was not able to access the supplementary figures by DOI. Furthermore, Figure 3 is too small and therefore not interpretable;
- in relation to the sampling estimate in the Materials and Methods section, bibliographies from much earlier years are cited (references 18 and 20);
- there are vague sentences, such as: "not all collected specimens were screened for influenza A because..." on page 5;
- the Discussion section could have a greater impact. Epidemiological and genetic characterization studies are important for the purpose of preventing the spread of infections. However, there is no mention of the fact that the presence of markets with live animals of different species that remain in contact for several days, could facilitate the species jump of infections. I suggest evaluating the results with a view to future improvement of infection prevention procedures;
- there are acronyms not explicitly stated in the text;
- there are some typing errors and parts of the text in another font.

Author Response

We thank the reviewer for taking the time to thoroughly review our manuscript and sincerely appreciate the valuable feedback and constructive comments, which have greatly contributed to strengthening the work.  We have carefully addressed all the comments and suggestions to ensure that our final submission meets the highest standards.   Below are responses to specific comments:-

The study presented is noteworthy, but suffers from the following critical issues:
Reviewer comment:  I was not able to access the supplementary figures by DOI.

Response: We regret that the figures could not be accessed through the DOI. To facilitate the review, we are uploading the supplementary figures with this submission. We also seek guidance from the editorial team if there is a step we might have missed with the DOI submission.

Reviewer comment: Furthermore, Figure 3 is too small and therefore not interpretable;

Response: We regret this challenge and have replaced this figure which is bigger and 

For figure 3- -

Reviewer comment: in relation to the sampling estimate in the Materials and Methods section, bibliographies from much earlier years are cited (references 18 and 20);

Response: We acknowledge the reviewer's regarding the earlier references cited.  However, we could like to highlight that reference 18 which reports results of systematic LBM surveillance  from 2009-2011 was utilized to calculate the sample size for our study based on reported prevalence and similar methods applied. In addition, we have included in the last paragraph of the introduction a more recent prevalence rate of 5.7% that was reported from a limited sample size (n=141) of poultry in an LBM in Kenya during January 2017. It is important to note that this particular paper had not been published at the time our study was designed, and we have made reference to if accordingly. Further, scarcity of recent citations on this topic from Kenya further emphasizes limited focus on AIV surveillance in this region.  

For reference 19, we acknowledge that there are more recent citations on sampling techniques. However, we chose to use the Thompson et al as they adopt a comprehensive approach to sample size calculation in animal populations specifically accounting for cluster sampling. This ensured robustness of out sample size determination. 

Reviewer comment:  there are vague sentences, such as: "not all collected specimens were screened for influenza A because..." on page 5;

Response: We have rephrased this sentence and now reads "Due to the high cost of reagents, a subset (72%) of collected specimens were screened for influenza A"
Reviewer comment:  the Discussion section could have a greater impact. Epidemiological and genetic characterization studies are important for the purpose of preventing the spread of infections. However, there is no mention of the fact that the presence of markets with live animals of different species that remain in contact for several days, could facilitate the species jump of infections. I suggest evaluating the results with a view to future improvement of infection prevention procedures;

Response: We thank the reviewer for this comment and observation and agree that a key message had been overlooked. We have included two statements in the discussion:

  1. In the second paragraph: "Additionally, the detection of AIVs in different poultry species that come into contact at LBMs for extended periods, up to 17 days, could potentially facilitate cross-species transmission".
  2. In the last paragraph: 'Adoption of suitable biosecurity measures in LBMs could mitigate cross-species transmission and opportunity for virus reassortments"

 Reviewer comment - there are acronyms not explicitly stated in the text; 

Response: We have included the following acronyms that had been omitted: ARI in the abstract;  CDC, GISAID, MUSCLE, HAU in the methods section, and RT PCR, CL and OP as footnote of Table 2

Reviewer comment: - there are some typing errors and parts of the text in another font.

Response: We have read through the manuscript and corrected the typing errors and applied the same font in the entire MS including text, tables and references. 

Reviewer 3 Report

Comments and Suggestions for Authors

The review on manuscript "Characterization of Avian Influenza Viruses Detected in Kenyan Live Bird Markets and Wild Bird Habitats Reveal Genetically Diverse Subtypes and High Proportion of A(H9N2), 2018- 2020”.

The peer-reviewed article describes the monitoring of avian influenza viruses in birds at live poultry markets, in poultry workers, and in wild migratory birds in Kenyan. Influenza virus was detected in 3.9% (n=292) of specimens collected from poultry.  Of 34 specimens randomly selected for further subtyping, all were H9N2. None of swabs from poultry workers were positive for influenza  virus. During the same period, a low-pathogenicity H5N2  and H11 viruses was detected in a fecal specimen of wild birds. The authors conclude that all H9N2 viruses isolated from poultry birds belonged to the G1 lineage, suggesting that viruses from a single lineage circulate.

Note:

1)      The text contains several technical errors, such as a different font.

(Examples:   In response to the outbreak risk assessment considered the spread of H5N8 across Eastern African countries as likely                       

Nairobi and one market in each of the other counties.    

Pooled specimens were tested

 were positive for influenza A segregated as:       

2)     Figure 3 is poorly readable. Details mentioned in the text are not visible. A PDF version should be attached.

Author Response

We thank the reviewer for thoroughly reviewing our manuscript which will improve the quality of our work. Below are the responses to specific comments: 

Reviewer comment: The text contains several technical errors, such as a different font.

(Examples:   In response to the outbreak risk assessment considered the spread of H5N8 across Eastern African countries as likely                       

Nairobi and one market in each of the other counties.    

Pooled specimens were tested

 were positive for influenza A segregated as:       

Response: We thank reviewer for catching this. We have applied the same font throughout the MS including the references. 

Reviewer comment:  Figure 3 is poorly readable. Details mentioned in the text are not visible. A PDF version should be attached.

Response: Indeed this important figure is pixilated. We have replaced this figure - which we hope has better resolution and readable. 

Round 2

Reviewer 2 Report

Comments and Suggestions for Authors

The revised version of the manuscript has been sufficiently improved/corrected.